# The influence of non-family members in top management teams on research and development investment: Evidence from Chinese family firms

Yujia Shao[1,2], Dechun Huang[1], Lelin Lv[1]*, Jie Yu[1]

1 Business School, Hohai University, Nanjing, China, 2 Business School, Wuxi Taihu University, Wuxi, China

* lvlelin1205@163.com

**Data Availability Statement:** All relevant data are within the manuscript and its Supporting Information files.

## Abstract

The diversified management ability of the non-family members in the top management teams (TMTs) can significantly increase the research and development (R&D) investment of the family firms. However, existing studies focus on family characteristics. To bridge the gap, this study explored the R&D investment propensity for family firms from the perspective of non-family members' participation in TMTs. Based on the upper echelons and the socioemotional wealth theory, this paper incorporated the non-economic goals that influence strategic decisions on family firms into the analytical framework. According to the questionnaire data of Chinese private enterprises, the Tobit regression model was used to analyze the influence of family members on R&D investment decisions under non-economic goal orientations. The results indicated that the preference for control and influence among family members weakens the positive effect of non-family managers on R&D investment, while the preferences for status perception and social responsibility strengthen the positive effect.

## 1. Introduction

With the rapid development of science and technology and the market demand for technology, the innovation capability is considered a key driving force for technological progress and economic growth in the current market. Although family firms make a significant contribution to the global economy and the technology sector, they are often portrayed as highly conservative and risk-averse, and the effect of family ownership on firm innovation has been also controversial [1, 2]. These characteristics have inspired a great deal of research on the attitudes of family firms toward research and development (R&D) investment [3–9].

Some studies have consistently concluded that family firms tend to invest less in innovation in order to maintain socioemotional wealth (SEW) based on prospect and agency theory [10–12]. However, there are still some scholars that support the opposite view. Specifically, family involvement in top management can effectively create distinct resources, which is regarded as the foundation for the subsequent improvement of firms' innovation ability [13–15]. In addition, as innovation becomes more and more important in the market, this trend will further

**Funding:** Financial support from Innovative Team of Philosophy and Social Sciences in Jiangsu Higher Learning Institutions, recipient H.D., grant numbers 2017ZSTD002. http://jyt.jiangsu.gov.cn/. The funders had no role in study design, data collection and analysis, decision to publish, or preparation of the manuscript.

**Competing interests:** The authors have declared that no competing interests exist.

encourage these family firms to invest in R&D [16]. Therefore, the findings on the relationships between the family involvement and R&D investments have been mixed.

Previous studies always focused on the impact of family on R&D investment [5, 6, 17]. In fact, whether to invest in R&D is a part of the business strategy, which largely depends on the decisions of the top management teams (TMTs) [18]. TMTs members are widely considered as some of the most crucial decision-making units in a firm, and organizational achievements are often recognized as a result of the efforts and capabilities of the TMTs members [19, 20]. Especially in family firms, the current research indicate that the reason that the family firms exploit significantly fewer opportunities than non-family firms is fully mediated by the organization of their TMTs [21]. Furthermore, as limited professional knowledge of family members and lack of resources, the innovation ability of the family firms may be affected. Given that most family firms are small and medium-sized enterprises (SMEs) [22], such firms often tend to rely on the non-family members of the TMTs who have expertise in specific fields of technology [23]. The non-family members in the TMTs, which consider as the resource of external R&D and technologies, play an important role in the R&D investment of the family firms [24, 25]. However, few of these non-family factors have been considered in the current literature, thereby leading to the inclusive results.

In contrast to previous studies that focused only on family influence, this study investigates the influence of non-family member managers of TMTs in the decision-making of R&D investment in family firms. Drawing on the upper echelons theory [26], this study argues that the family firms undertake a higher level of R&D investment when non-family members participate in TMTs at high levels. A central premise of the theory is that the involvement of non-family members in TMTs brings in more diversities to management, which benefits the family firms to make accurate judgments in identifying or pursuing creative and innovative opportunities [27]. The non-family members in TMTs are different from family members, who are motivated by more than just financial goals. Accordingly, the non-family members tend to better facilitate the firms to seize key opportunities during the market competition [28]. In addition, Vardaman and Gondo [29] have argued that the preferences of different SEW dimensions may lead to conflicts with the strategic decisions of senior executives within the firms. SEW as a reference point for family business decision-making reflects more non-economic goals, but such non-economic goals are heterogeneous in different family businesses and even in different environments [30, 31]. Therefore, the relationship between the involvement of non-family members in R&D activities and R&D investment in family firms is contingent on the concerns of the family owners' non-economic goals.

As China transitiones from a centrally planned economy to a socialist market one, the family firms have occupied the main share [32]. Innovation is of great significance to Chinese family businesses against the background of accelerating economic transformation [33]. In response to the need for market expansion, it has become progressively obvious that family leaders adopt the strategy of de-familization by introducing talents and funds [34]. With the involvement of non-family managers in TMTs, the family influence in strategic decision-making wanes out, and the non-family members start getting a greater say in decision making. Moreover, because the Confucian culture strongly promotes the value of family ties in maintaining group solidarity and social order [35], the preference for non-economic goals among Chinese family firms may be different from the dimensions proposed by earlier scholars [36]. Therefore, this paper analyzes the influence of family members on R&D investment decisions under different non-economic goal orientations based on SEW theory and Tobit regression model, taking Chinese family firms in a specific cultural environment as the research object.

The main contributions of this study are as follows: First, most of the previous studies explored the influence of TMTs on innovation-related decision-making from the perspective

of family involvement. On the contrary, this study analyzes the impact of non-family members' participation in TMTs on organizational innovation decisions from the perspective of their participation. Second, based on the traditional Chinese culture and the managerial concepts of family firms [37], this paper introduces the non-economic goals of family members (control and influence, status perception, and social responsibility), and comprehensively analyzes the joint effects of non-family factors and family factors on the R&D investment decision-making of family firms. Lastly, previous studies have focused more on family businesses in Europe but less on those in developing countries. This study can provide enlightenment and reference for enterprises worldwide through the studying of a few Chinese family firms.

## 2. Theoretical background

### 2.1 Top management teams and research and development

The R&D investment for developing new products is closely related to the competitiveness of the enterprises in the market [38]. As an important strategy for a company, innovation decisions cannot be implemented without the support of top management. Based on the upper echelons theory, strategic decision-making reflects the thoughts of the TMTs. Therefore, when top executives possess heterogeneous thought processes, values, and perceptions, the firm's strategic choices and subsequent performance outcomes will benefit [39]. In view of this, some scholars suggest that other TMT-related factors also affect innovation. Tuncdogan et al. [40] found that the regulatory focus of a TMT could promote exploratory innovation significantly. Li et al. [41] argued that the task-related diversity of TMTs represents the differentiating cognition for multifarious innovation, thus accelerating the speed of innovation.

It is to be noted that most family firms in China are middle and small-sized enterprises. Premkumar et al. [42] proposed that SMEs and large enterprises have differences in decision-making ways. Large enterprises typically have a chief officer for decision-making, while SMEs must rely on the collective knowledge and experience of the TMTs [43]. Especially, family firms are prone to be restricted by factors such as lack of understanding of the market demand, limited knowledge in technology development, and the appeals to the self-interest of the members, which makes it difficult for family leaders to formulate innovation strategies for the firms. Therefore, the introduction of external managers into TMTs has become a common phenomenon in enterprises [24]. Although some previous studies have focused on the relationship between the TMTs in family firms and the R&D, these studies analyzed the relationship from the perspective of family involvement, such as the generational involvement in TMTs and family members in TMTs [44, 45]. However, little is known about the influence of non-family members on the strategic decisions of family firms. Therefore, it is necessary to explore the influence of non-family members in TMTs on the R&D investment of family firms according to the current development trend of the Chinese family firms.

### 2.2 Socioemotional wealth theory

With an increasing number of research studies on family firms, Gómez-Mejía et al. [12] developed the SEW theory to provide a reasonable explanation for the heterogeneous behaviors of family firms. The core idea of this theory suggests that firm-owning families pursue control of the businesses in order to satisfy their effective desire for authority, influence, and internal identity. The SEW theory explains why the owners of family firms always tend to make decisions that are partial to non-economic interests, and more conservative than leaders in non-family firms [46]. It is widely believed that SEW plays an important role in the strategic decision-making of the family firms [31, 47].

The extant literature emphasized that SEW has different dimensions such as family control and influence, family members identification, binding social ties, emotional attachment, and the renewal of family bonds to the firms through dynastic succession (FIBER) [36]. Similarly, Miller et al. [46] also suggested that the goals of SEW are related to family motivation, and they divided the SEW into restricted and extended parts. However, arguments based on the SEW theory are still inconclusive while studying innovative investment behaviors among family firms [4, 9]. An important reason for this phenomenon is that the existing research rarely combines this theory with regional environment and generally overlooks the complex variations among the family-run firms [48]. Specifically, SEW can be considered a kind of emotional preference among family members, and it embodies the values of family members. Since personal values is closely related to the surrounding culture and environment, it is inappropriate to use these dimensions proposed by previous scholars to analyze the impact on innovation directly. Furthermore, Chua et al. [49] questioned the accuracy of SEW and proposed that the relationships among the various SEW dimensions may be more complex than theorized in the current literature. Jiang et al. [50] also believed that socioemotional goals might be driven by distinctive factors. Moreover, during the decision-making process of a family firm, the various SEW dimensions might be given different priorities [49].

In summary, it is necessary to comprehensively consider the influence of non-family factors and family goals on innovation decisions of the family firms. What's more, when it comes to family goals, the research cannot only be conducted according to the classification of previous SEW. It is necessary to comprehensively analyze the heterogeneous behavior decision-making and innovation of family firms by considering the external variables (e.g. the regional humanistic environment) and internal variables (e.g. the traits of character). Therefore, based on the SEW theory and the emotional preferences of Chinese entrepreneurs, this paper explores the influence of non-family factors on firm R&D investment by combining the preferences of three emotional goals (control and influence, status perception, and social responsibility) of family firms. Specifically, once the family members lose the power of control, the family firms cease to be possessed family traits. Therefore, the primary goal of the family members is to maintain absolute control. Furthermore, many family firms in China have started from the grassroots, and through their efforts broke down the social hierarchy. In this context, family members are extremely sensitive to external recognition for their own identity. This preference of identity, as well as non-economic interests, also influences organizational behaviors [51]. Therefore, the perception of a leader's identity is also seen as an emotional goal. Finally, the development of family firms is closely related to the interests of the family members, wherein the emotional preferences are based on the joint efforts of the stakeholders and the public recognition. As a result, the emotional preferences of family firms are contingent on the joint efforts of the stakeholders and the recognition of the public [52]. Also, the family members are more likely to regard the firms as an extension of themselves and show greater social responsibility.

## 3. Hypotheses

Based on the characteristics of Chinese family firms, this study examined the impact of importing outsiders into TMTs on the strategic management and innovation goal intention of family firms. In addition, considering the influence of family members on management decisions, this study divided the affective goals of family members into three dimensions: control and influence, status perception, and social responsibility. It also analyzed the moderating effects of these emotional goals on the relationship between the involvement of non-family members

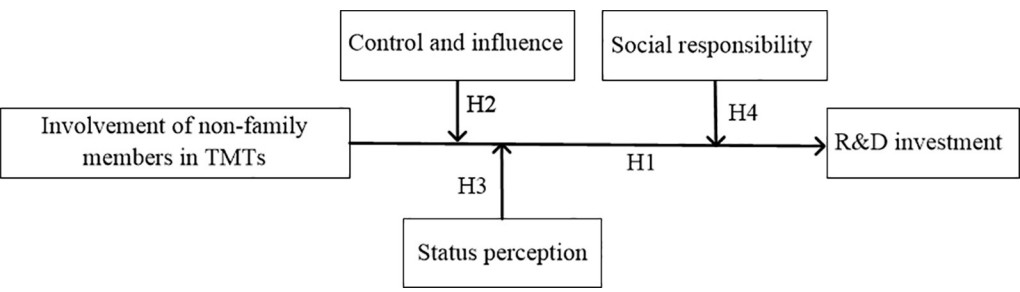

**Fig 1. The conceptual framework of hypotheses.**

in TMTs and R&D investment. Fig 1 shows a visual representation of the core constructs in our study and the proposed relationships among them.

## 3.1 Involvement of non-family members in TMTs and R&D

The involvement of non-family members in TMTs indicates an inclusive mindset of the family members. Firms with diversified TMTs need to the consideration of many alternatives, which increases the likelihood of innovative decisions. The existing literature suggests that top managers with diverse educational and organizational backgrounds can combine different views of the world and have more constructive task conflicts, which results in the firm promoting a proactive innovation orientation [18, 53]. In addition, compared with firms without diversified management teams, groups that possess diverse knowledge and perspectives tend to incur conflicting opinions in some important strategic decisions. Therefore, it is easy for such diversified organizations to improve the decision quality [54], and family members are also found to support these intelligent decisions with respect to innovation.

At the same time, the involvement of non-family members in TMTs enhances the ability of the firms to create continuous value and expedite knowledge transfer [55]. Specifically, the non-family members bringing in many social networks and firm-specific, tacit knowledge, provide a new channel for the family members to access new perspectives and ideas. What's more, non-family executives can reduce the tendency toward convergent thinking among family firm managers [56]. Hence, it becomes easier for these firms to break down the stereotypes that tend to avoid investing in risky innovations. As the non-family executives do not represent the vested family interests, they could give suitable suggestions based on rational analyses about the innovation strategy, unlike the family members who are restricted by the family emotions [57]. The diverse and rich professional knowledge from non-family TMTs members can increase the identification and pursuit of creative and innovative opportunities for the firms [27].

Although non-family members have their advantages in team management, some existing studies argued that there are more conflicting relationships between team management teams composed of mixed family members and non-family members [49, 58]. Based on the agency theory, the non-family members may conduct opportunistic behavior when they suffer injustice [59]. However, this aspect may be solved well by most Chinese SMEs. In China, most private entrepreneurs tend to pass on their firms to the next generation, thus maintaining better family management [60, 61]. Especially, Chinese family firms are entering the important time of succession currently [62]. Therefore, the Chinese traditional patriarchal view tends to eliminate the obstacles and the disadvantages of the existing firms so that the next generation can successfully inherit the family firms and maintain control of it successfully [63]. Furthermore, the younger generations are hard to get support and recognition from the TMTs, as the huge

influence of and the absolute authority created by the older generations do not easily get transferred to their successors. In this context, strategic changes could help break the conventions. Compared to older generations, the younger generations have more adventurous and newer visions. They want to establish their prestige by making great achievements and prefer projects with risky prospects, such as investing in innovation activities. The older generations also take some measures, such as coordinating the relationships between the family members and the non-family members in the TMTs, by which they could inject vitality and lay the foundation for the long-term development of the enterprises. These measures also help create a favorable business environment for the next generation to succeed smoothly. The notion of innovation ability matters for family intergenerational transition. It resonates with the specialized knowledge and ideas from non-family executives, gives non-family members a greater say in the affairs, and increases trust in the family members [64]. Therefore, the following hypothesis is proposed:

**Hypothesis 1**: *The involvement of non-family members in the TMTs is positively related to the R&D investment of a family firm.*

## 3.2 Moderating role of family control

The main distinction between family and non-family firms lies in whether some important strategic decisions are made by a few large shareholders with family connections [65]. In other words, it is difficult for non-family members to have the power to make key decisions in family firms. As the preference of the family members about the influence and control deepens, it is common for family elders to designate family members in key positions and satisfy the family's desire for continuity by using their privilege regardless of financial considerations [12]. Such a strong emotion reinforces the need of the family members for controlling the business, which increases the possibility that the enterprise deviates from the normal operational track.

From the perspective of this emotional goal of enterprise control and influence, two kinds of cases can occur in the firms inhibiting innovation investment. Firstly, investment in R&D activities is sunk cost and need continued financial support [66], and it is usually difficult to further invest only using the family's own capital. However, family firms are labeled with an image that is reluctant to increase their debt levels or to raise money from the stock market, because the introduction of external investors implies the dilution of control of existing family members, causing their goals to be at risk. In this situation, family members tend to avoid deciding on investing in innovation [63]. Secondly, the contribution of non-family executives is easily weakened to ensure the influence of family members in the enterprise, leading to a sense of injustice among the non-family managers [67]. Particularly, R&D activities have huge risks and long cycles, non-family executives are likely to become scapegoats when an activity fails, so non-family executives do not actively promote innovation investment. Therefore, the following hypothesis is proposed:

**Hypothesis 2**: *The preference for control and influence among family members weakens the positive relationship between the involvement of non-family members in TMTs and the family firms' R&D investments.*

## 3.3 Moderating role of status perception among family members

For family members, non-economic goals are a kind of psychological-emotional endowments, such as the control and influence on the internal firms. Similarly, the perception of one's own identity or status is also the focus of the psychological needs of the family members. Berrone

et al. [36] argued that the identity of the family leader is inseparable from the family name carried by the firms. In other words, the social status of the organization is closely related to the social status of the family members and is rooted in the deep psychological level of the family owners [68]. Since the reform and opening up, the social influence of private entrepreneurs has been continuously improved, so that their social identity and status have been rising with the growth of enterprises. Importantly, objective measures of social identity are mostly related to people's subjective perceptions of their status. In family firms, the emotional wealth about the perception of social identity among the principals is considered to be a non-financial factor and represents the pivotal frame of reference in strategic decision-making [69]. Especially, the most direct impact of such status promotion on the firms is the change of strategic direction. Some literature about psychology, education, and sociology suggests that social classes can have a profound influence on individuals' perspectives and their decision-making [51, 70]. This view is also supported by the upper echelons theory, which holds that managers of different social classes tend to have very different strategic orientations [71].

According to some previous studies, the perception of an individual's social status has two important characteristics: embeddedness and comparability. The former is to integrate oneself into a specific group, and the latter is to compare oneself with the individuals in the embedded group [72]. In China, the status perception of entrepreneurs mainly consists of three factors: whether they are recognized by the government, their income status, and the public's recognition of their professional reputation [73, 74]. Therefore, it is necessary to analyze the influence of non-family members on R&D investment in TMTs of family firms through the self-evaluation and perception of family firms' leaders on their economic, political, and social status in combination with Chinese social conditions.

Leaders of family firms with a high sense of identity tend to be better able to take risks. On the economic level, the abundant capital provides the entrepreneurs with a kind of psychological security, even if some damage or failure happens in the process of innovation, and they still can recover. This mentality leads to higher levels of optimism and confidence, which may make them underestimate the potential losses in certain situations and even encourage them to pursue such adventurous activities [75]. Especially during the transition period of social and economic changes, Chinese private entrepreneurs have been facing huge mental and physical pressure from the institutional reform and market competition, yet higher subjective perceptions enable them to minimize the adverse effects. Based on the current Chinese social context, the awareness of the social and political status of the entrepreneurs largely depends on their links with the government and the social evaluation, so these entrepreneurs can benefit from a series of privileges by the government or some external conditions such as enterprise R&D subsidy when the family leaders have a high level of social and political status [76, 77]. Moreover, in recent years, as the Chinese government attaches great importance to the innovation ability of private enterprises, local governments have been increasing their support for the innovation activities of family firms. It is easier for enterprises to obtain social recognition and support from local governments when implementing innovation strategies. Therefore, the following hypothesis is proposed:

**Hypothesis 3**: *The perception status of family members strengthens the positive relationship between the involvement of non-family members in TMTs and the family firm's R&D investment.*

## 3.4 Moderating role of social responsibility among family members

In the existing literature, SEW has been described as an act with a selfish nature and largely meets the personal affective desire of family members [78, 79]. However, it is too narrow a

view, where the reference point for family interest only in decision making could largely be irrational, and maintaining the normal operations of the enterprises could be difficult. Although the family members have motives related to their own personal interests such as permanent job security or the control of and influence on the firms, as proposed by Miller and Le Breton-Miller [46], yet another type of SEW, called the extended SEW can be noticed as well. Accordingly, the family not only pursues its own interests but also serves the interests of the stakeholders, thereby increasing the ability of the firms to run in the long term [80]. In fact, the family firms in China would not have developed so well if the family members only pursued their personal interests. Therefore, it is necessary for us to further understand the non-economic goals of another level of Chinese family firms through the extended SEW. The extended SEW is in line with the idea mentioned by Newbert and Craig [78] that family firms consider their own interests and those of others to be complementary, rather than competing, and provide a strong basis for decision-making. In this context, it is conducive to maintaining the stability of the market, effectively ensuring the long-term development of the family firms so that family members are more likely to view the firms as an extension of themselves [81]. In general, family members are more inclined to avoid situations that may give a negative impression to their organization, which forces them to engage in social responsibility for maintaining a good reputation. Therefore, the behaviors related to corporate social responsibility (CSR), such as protecting the interests of the stakeholders and the employees, improving the waste management process, could be considered as a way to protect the external image.

In fact, CSR is seen as an additional strategy to maintain or increase profitability and boost development [82]. With the increasing complexity of business and technological change, competitiveness increases significantly. Therefore, the firms that practice social responsibility consider CSR as an important issue on the corporate goal and strategy making. Many scholars have studied the relationship between social responsibility and organizational behaviors, especially innovation activity [83]. Similarly, as mentioned above, CSR is one of the goals of family firms. Family members' concern for corporate social responsibility can also influence the innovation decisions made by the TMTs.

From the perspective of stakeholders and the Chinese market environment, this paper argued that the practice preference of social responsibility will actively promote innovation decision-making under the mixed organizational structure of family members and non-family members. Firstly, CSR reflects the inclusiveness and the sense of responsibility of firms, indicating that the development of the firm is not only oriented by the interests of the family members. Socially responsible firms are more likely to enjoy greater trust, loyalty, and higher levels of satisfaction from the various stakeholders, including the customers, employees, investors, business partners, and the communities [84, 85]. Compared to the family members, the non-family executives are external stakeholders of firms. As noted above, this goal preference is more likely to help non-family members develop a sense of emotional security because their interests are effectively protected and even creates an incentive for them. In addition, the corporate social responsibility projects can enable firms to build a broader and deeper network of relationships with the various stakeholders, and promote value sharing and exchange among external knowledge stakeholders, which is important for the innovation strategy of firms [86, 87]. Secondly, under the market orientation of advocating mass innovation, family firms tend to cater to government policies and actively build a good corporate image to get more support from the government [88]. What's more, the economic growth has gradually entered a new normal along with the transformation of the Chinese economic development model. CSR helps firms absorb high-quality talent and information resources in an environment of uncertain government policies, and motivates family firms to engage in high-risk activities such as innovation investment [89]. Therefore, the following hypothesis is proposed:

*Hypothesis 4*: *With a sense of social responsibility, the family members strengthen the positive relationship between the involvement of non-family members in the TMTs and the family firm's R&D investment.*

## 4. Methodology

### 4.1 Sample and procedures

The sample was drawn from the sample survey database of Chinese private enterprises in 2010. The sample survey database included private enterprises of different sizes from various industries in 31 provinces, autonomous regions, and municipalities in China, and conducted a nationwide multi-stage sampling of enterprises with a proportion of 0.55%. A total of 4,900 questionnaires were sent out and 4,614 were collected with a total recovery rate of 94.16%. The names of the institutional review board that approved the questionnaire survey are the Institute of Sociology Chinese Academy of Social Sciences and All-China Federation of Industry and Commerce, which issued a statement approving this research (http://ssm.cssn.cn/sjzy/201609/t20160914_3202846.shtml). Moreover, the survey data was analyzed anonymously. The focus of the study is on owners of family firms, so in this study, enterprises with a family shareholding ratio greater than or equal to 50% were selected as the samples of family firms with absolute family control [90]. Those firms with unreasonable or missing research are deleted. After filtering according to these conditions, our analysis sample consisted of 1,098 family firms.

### 4.2 Measurement

**4.2.1 Dependent variable.** The dependent variable measures the intention of the firms in investing in innovation. According to Duran et al. [1], R&D intensity is used to represent the proportion of R&D expenditure in the sales revenue of a firm. That is, the proportion of R&D expenditure in the sales revenue of the enterprise in the returned questionnaire; Call it R&D. That is, the proportion of the R&D expenditure in the returned questionnaire to the sales revenue of the enterprise, which is called the R&D.

**4.2.2 Independent variable.** The focus of the study is on the impact of non-family members in TMTs on innovation investment in family firms. The power of decision-making is controlled by the board of directors, so the independent variable is measured by calculating the percentage of non-family members in the total members of the firm's board of directors, namely the Non-family power (Nfp) [59].

**4.2.3 Moderating variables.** This paper measures the control preference and influence of family members from three perspectives of family non-economic goals, namely, ownership control, strategic control, and management control. There are three main problems: (1) Whether the family ownership is greater than 50%; (2) Whether the strategic decisions of the firms are controlled by the family members; and (3) Whether the key positions in the firms are held by the family. These questions were measured on 5-point Likert scale ranging from one (strongly disagree) to five (strongly agree). The Cronbach's alpha of these questions is 0.84.

Another perspective of goal is measured by the status perception of the family leaders. The status itself is a matter of subjective perception and evaluation, and sometimes subjective status has a greater impact on individual psychology and behavior than objective status. According to Ma et al. [72], combining three dimensions of social stratification, such as economic status (wealth and income), political status (power), and social status (prestige), the interviewed were asked the following questions: (1) Compared to the other members of the society around you, where do you think you are on the following three social ladders in economic status? (2) Compared to the other members of the society around you, where do you think you are on the

following three social ladders in political status? (3) Compared to the other members of the society around you, where do you think you are on the following three social ladders in social status? For each ladder, the response must be a number between 1 and 10, where 10 represents the highest point, and 1 represents the lowest point. The Cronbach's alpha of these statements is 0.91.

Since corporate social responsibility could reflect in many ways, it is impossible to measure every aspect separately. According to the current behavior trend of the Chinese family firms in practicing social responsibility [91, 92], this paper measures four aspects: environmental impact, social welfare, employee relations, and production safety. In the questionnaire, there are four corresponding survey questions, with binary options, 0 (for No) and 1 (for Yes): (1) Whether to invest in pollution control; (2) Whether the firm has donated to public welfare undertakings in the past two years; (3) Whether the firm receives government subsidies for recruiting people who are feeling difficult to find a job; and (4) Whether to invest in safety equipment.

**4.2.4 Control variables.** Some variables that have been shown to influence the R&D investment are included, which are divided into two parts. The first is the characteristics of the family leader, who is the most authoritative person and whose judgment exerts a key influence on the strategy in a firm. According to the upper echelons theory, the strategic decision of a firm is an embodiment of the cognition and thought of decision-makers. Furthermore, the education level, gender, age, the social identity of the senior executives determine their cognitive perspective and thought [93]. Thus, the leader's age, education, gender, and political connections were examined. Education is measured by the educational background of the leader, and it is graded from low to high: elementary school, junior high school, senior high school, technical secondary school, undergraduate, and postgraduate (The responses must be numeric from 1 to 6, where 1 represent elementary school, and 6 represent the postgraduate) [94]. What's more, the male is represented by 1 and the female by 0. Similarly, if the leader has a government job, it is coded as 1, otherwise as 0.

The other part is the characteristics of the firm. The importance of factors that can influence innovation in family firms by several specific control variables (firm scale, firm age, the rate of profit rate, the revenue growth rate, the proportion of family shareholding). The firm scale is measured by the number of employees, because the firm scale is closely related to the level of knowledge acquisition, and which could influence investment in innovation [95]. The firm age is calculated based on the difference between the year (2009) in which the survey was conducted and the year in which the firm was founded. Last year's profit margin and revenue growth rate were selected as control variables because R&D investment involves risk, which may be affected by last year's performance. The higher the proportion of family shareholding is, the greater the loss brought by operational risk to the family will be, which will further affect the strategic decisions related to innovation [96]. Therefore, the proportion of family shareholding, as expressed by ownership is controlled.

## 4.3 Estimate model

The dependent variable is a continuous variable, limited to values between 0 and 1. Therefore, the Tobit regression model is the most appropriate statistical operational model for continuous innovation measures. The model can avoid inconsistencies or biases, which are more likely to occur in the estimation of these variable types [97]. In this study, the Tobit regression model can be expressed as follows:

$$R\&D = \beta_0 + \beta_1 Nfp + \beta_2 Controls + \varepsilon \tag{1}$$

To explore the moderating effects of control and influence, perception status, and social responsibility, this paper mean-centered the variables and created the two-way interaction terms by multiplying the two centered variables together. Hence, these interaction terms (Nfp*Control, Nfp*Status, Nfp*CRS) were added to Eq (1). Accordingly, the estimation model is as follows:

$$R\&D = \beta_0 + \beta_1 Nfp + \beta_2 Moderate + \beta_3 Nfp*Moderate + \beta_4 Controls + \varepsilon \qquad (2)$$

Where *R&D* represent R&D intensity, $\beta_0$, $\beta_1$, $\beta_2$, and $\beta_4$ are the estimated coefficients. *Moderate* includes Control, Status, and CRS. 03b5 $\varepsilon$ is an exogenous random variable.

## 5. Data analysis and results

### 5.1 Descriptive statistics

The data collected was analyzed using STATA 14.0 software. The descriptive statistics of all variables for the family firms were present in Table 1. The average age of these firms is 10 years, and each firm has approximately 221 employees. In terms of R&D intensity, most firms are at a lower level (Mean is 0.012), which reflects the weak innovation ability of Chinese private enterprises. The average gender score for the family leaders is 0.864, indicating that most firms are controlled by males. From the perspective of the correlation among the variables, the R&D intensity of the firm is significantly related to the Nfp variable ($r = 0.058$, $p<0.05$), Non-financial goals for control and influence ($r = -0.06$, $p<0.05$), perception of status ($r = 0.032$, $p<0.01$), and social responsibility ($r = 0.103$, $p<0.01$). This primarily verified the hypotheses of this study. These correlations indicate that no issues exist with multicollinearity among the

**Table 1. Descriptive statistics and correlation.**

| Variable | 1. | 2 | 3 | 4 | 5 | 6 | 7 | 8 | 9 | 10 | 11 | 12 | 13 | 14 |
|---|---|---|---|---|---|---|---|---|---|---|---|---|---|---|
| 1. R&D | 1 | | | | | | | | | | | | | |
| 2. Nfp | 0.058** | 1 | | | | | | | | | | | | |
| 3. Control | -0.06** | -0.250*** | 1 | | | | | | | | | | | |
| 4. Status | 0.032*** | -0.006 | -0.013 | 1 | | | | | | | | | | |
| 5. CRS | 0.103*** | -0.017 | -0.052 | 0.132*** | 1 | | | | | | | | | |
| 6. Gender | 0.062 | 0.035 | 0.023 | 0.074 | 0.123** | 1 | | | | | | | | |
| 7. Education | 0.156*** | 0.106** | -0.106** | 0.076 | 0.004 | 0.033 | 1 | | | | | | | |
| 8. Politics | 0.026 | 0.068 | -0.096** | 0.264*** | 0.116** | 0.21*** | 0.11** | 1 | | | | | | |
| 9. Ownership | -0.004 | -0.481*** | 0.124** | 0.07 | -0.063 | -0.001 | -0.016 | 0.057 | 1 | | | | | |
| 10. Sale | 0.045 | 0.035 | -0.049 | 0.071 | -0.011 | 0.087* | 0.082* | 0.029 | 0.024 | 1 | | | | |
| 11. Profit | 0.078 | 0.048 | -0.013 | -0.024 | -0.047 | 0.083* | 0.048 | 0.034 | -0.003 | 0.414*** | 1 | | | |
| 12. Age | 0.074 | 0.007 | 0.086* | 0.112** | 0.073 | 0.111** | -0.109** | 0.122** | -0.064 | -0.042 | 0.016 | 1 | | |
| 13. F_age | 0.07 | -0.093* | 0.041 | 0.137*** | 0.134*** | 0.149*** | 0.031 | 0.213*** | 0.029 | -0.081* | -0.025 | 0.237*** | 1 | |
| 14. Scale | 0.072 | -0.049 | -0.124** | 0.236*** | 0.355*** | 0.12** | 0.155*** | 0.321*** | 0.015 | -0.039 | -0.04 | 0.125** | 0.212*** | 1 |
| Mean | 0.012 | 0.595 | 2.845 | 4.966 | 0.517 | 0.864 | 4.229 | 0.64 | 0.856 | 0.196 | 0.285 | 46.8 | 9.926 | 221.7 |
| Std. Dev. | 0.02 | 0.349 | 0.947 | 1.594 | 0.197 | 0.343 | 1.059 | 0.48 | 0.185 | 0.391 | 0.822 | 7.68 | 4.23 | 251.69 |

Note: R&D represent R&D intensity.

* represents $p<0.1$.

** represents $p<0.05$.

*** represents $p<0.01$.

Nfp is the proportion of non-family members in the TMTs. Control is the wiling of a family member in the control and influence. Status is the perception Status of family members. CRS is corporate social responsibility. Politics is the political connection of a family leader. F_age is the age of the firms.

variables because the correlations are all below the threshold value of 0.65 [98]. Furthermore, the variance inflation factor (VIF) values associated with these variables ranged from 2.54 to 4.66, all of which were far below the acceptable upper limit of 10. Therefore, these results showed that multicollinearity is not critical.

## 5.2 Results analysis

The results of the regression were present in Table 2. Model 1 only includes control variables. The coefficient of ownership is -0.0022 ($p<0.05$), which showed the proportion of family shareholding is negatively correlated with R&D intensity, that is, the family firms are risk-aversive. In the meanwhile, Education ($r = 0.0048$) and Scale ($r = 0.0042$) are significantly and positively correlated with R&D intensity, indicating that the larger the enterprise is, the higher the education level of the entrepreneur is, and the more the firms are inclined to invest in R&D. Then, the independent variable is added into the Model 2. The coefficient of Nfp is 0.0041,

**Table 2. Results of Tobit regression analyses for R&D intensity of firms.**

| Model / Variable | Model 1 | Model 2 | Model 3 | Model 4 | Model 5 | Model 6 | Model 7 | Model 8 |
|---|---|---|---|---|---|---|---|---|
| Nfp | | 0.0041** (2.32) | 0.0038** (2.05) | 0.0038* (1.86) | 0.0050* (1.73) | 0.0034** (1.99) | 0.0060* (1.83) | 0.0074* (1.76) |
| Control | | | -0.0020** (-2.17) | -0.0010* (-1.81) | | | | |
| Nfp*Control | | | | -0.0035** (-2.23) | | | | |
| Status | | | | | 0.0004** (2.36) | -0.0008* (-1.88) | | |
| Nfp*Status | | | | | | 0.0014* (1.69) | | |
| CRS | | | | | | | 0.0351*** (2.69) | 0.0135** (2.36) |
| Nfp*CRS | | | | | | | | 0.0063* (1.88) |
| Gender | 0.0048* (1.82) | 0.0021 (1.06) | 0.0059 (1.49) | 0.0021 (1.23) | 0.0056 (0.98) | 0.0021 (0.93) | 0.0041 (0.86) | 0.0017 (0.94) |
| Education | 0.0042*** (3.12) | 0.0028*** (2.95) | 0.0045** (2.44) | 0.0027** (2.38) | 0.0045*** (2.55) | 0.0028*** (3.67) | 0.0050** (2.33) | 0.0029* (1.86) |
| Politics | -0.0031 (-0.48) | -0.0017 (-0.68) | -0.0023 (-0.76) | -0.0019 (-0.89) | -0.0023 (-0.64) | -0.0017 (-0.52) | -0.0019 (-0.96) | -0.0017 (-0.76) |
| Ownership | -0.0022** (-2.28) | -0.0048* (-1.83) | -0.0059 (-0.98) | -0.0050 (-0.60) | -0.0054 (-0.98) | -0.0047 (-0.65) | -0.0091 (-0.98) | -0.0057 (-0.63) |
| Sale | -0.0062* (-1.83) | 0.0025 (0.77) | -0.0033 (-0.99) | 0.0064 (0.74) | -0.0072 (-1.46) | 0.0095 (1.51) | -0.0025 (-0.88) | 0.0083 (0.74) |
| Profit | 0.0023 (0.95) | 0.0015* (1.73) | 0.0034* (1.85) | 0.0026 (1.23) | 0.0031* (1.71) | 0.0022 (0.83) | 0.0038* (1.85) | 0.0017 (1.13) |
| Age | 0.0108 (0.99) | 0.0213* (1.74) | 0.0199** (2.22) | 0.0246 (0.61) | 0.0183* (1.78) | 0.0190 (1.11) | 0.0194* (1.76) | 0.0193 (1.50) |
| F_age | 0.0019 (0.27) | 0.0023 (0.24) | 0.0031 (0.40) | 0.0024 (0.26) | 0.0028 (0.42) | 0.0023 (0.27) | 0.0022 (0.41) | 0.0020 (0.28) |
| Scale | 0.0048*** (2.91) | 0.0061 (1.08) | 0.0032*** (3.13) | 0.0105 (0.81) | 0.0033*** (2.83) | 0.0162 (0.81) | 0.0015 (0.42) | 0.0102 (0.85) |
| Cons | -0.9214*** (-2.83) | -0.5211*** (-2.45) | -0.1230*** (-4.02) | -0.0551*** (-3.4) | -0.1248*** (2.74) | -0.0465* (-5.38) | -0.1405*** (4.04) | -0.0574*** (2.504) |
| Log likelihood | 1221.423 | 1050.788 | 1154.366 | 1051.962 | 1342.791 | 1086.145 | 1193.784 | 1043.634 |
| LR chi2(9) | 82.13 | 19.89 | 22.83 | 32.24 | 20.67 | 30.61 | 23.66 | 29.59 |
| N | 1098 | 1098 | 1098 | 1098 | 1098 | 1098 | 1098 | 1098 |
| Left | 578 | 578 | 578 | 578 | 578 | 578 | 578 | 578 |
| Right | 520 | 520 | 520 | 520 | 520 | 520 | 520 | 520 |

Note: The Z-statistic is shown in parentheses.

* represents $p<0.1$.

** represents $p<0.05$.

*** represents $p<0.01$.

Scale and Age are taken by the natural logarithm.

significantly positive ($p<0.05$), which indicated that the investment in R&D increases with the involvement of non-family members in TMTs, and Hypothesis 1 is verified.

Models 3 to 8 were created to verify the moderating effect. Hypothesis 2 assumes that the preferences for control and influence among family members moderate the relationship between non-family members in TMTs and the innovation investment. According to Table 2, it can be concluded that the coefficient of family control and influence in model 3 is significantly negative ($\beta = -0.0020$, $p<0.05$). Moreover, the coefficient of control and the interaction term (*Nfp\*Control*) are all significantly negative ($\beta = -0.0010$, $p<0.1$; $\beta = -0.0035$, $p<0.05$). As a result, the preferences for control and influence of the family members negatively influence the relationship between non-family members in TMTs and innovation intensity, thereby supporting Hypothesis 2.

Similarly, in Models 5 and 6, this study examined the moderating effect of the status perception on the relationship between non-family members in TMTs and the R&D intensity. As it shows in Table 2, the moderating variable and the interaction term (*Nfp\*Status*) exert a positive impact on the innovation intensity ($\beta = 0.0004$, $p<0.05$; $\beta = 0.0014$, $p<0.1$). This result reveals that the positive correlation between non-family members in TMTs and innovation intensity strengthens with an increase in the status perception of the family leaders, which supports Hypothesis 3.

In terms of the social responsibility of family firms, the proposition is that those family members with a strong sense of social responsibility may moderate the relationship between non-family members in TMTs and the innovation intensity positively. According to the coefficient of the moderating variable and the interaction term in Model 7 and Model 8 ($\beta = 0.0351$, $p<0.01$; $\beta = 0.0063$, $p<0.1$), Hypothesis 4 is verified.

According to the data analysis results in Table 2, this paper illustrated the moderating effect of the family control and influence, the status perceptions of the family members, and the family social responsibility on the relationship between non-family members in TMTs and the innovation tendency of the firms. Fig 2 showed a lower slope for strong family control and influence (High Control) than for weak family involvement (Low Control). Fig 3 showed a

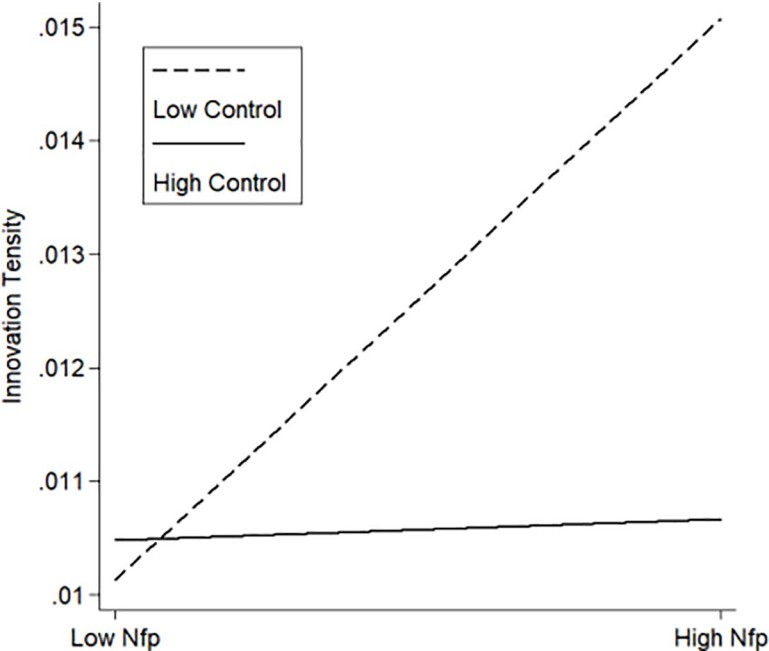

**Fig 2. The moderating effect of the family control and influence.**

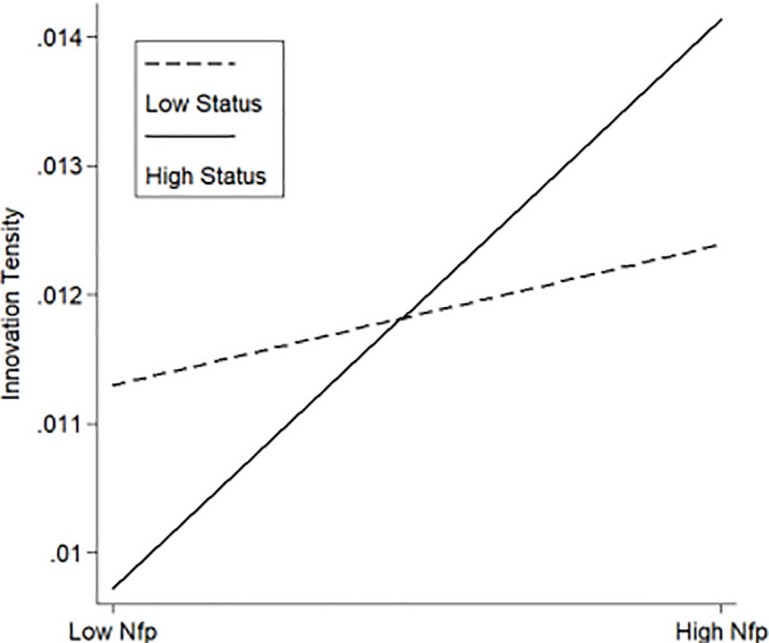

**Fig 3. The moderating effect of the status perceptions of the family members.**

steeper slope for strong family status perception (High Status) than for weak status perception (Low Status). A similar result was shown in Fig 4 in terms of social responsibility.

## 5.3 Robustness check

Although the above empirical analysis well verified the hypothesis proposed, it is observed that non-family board members only hold the shares and want to gain profit rather than engage in

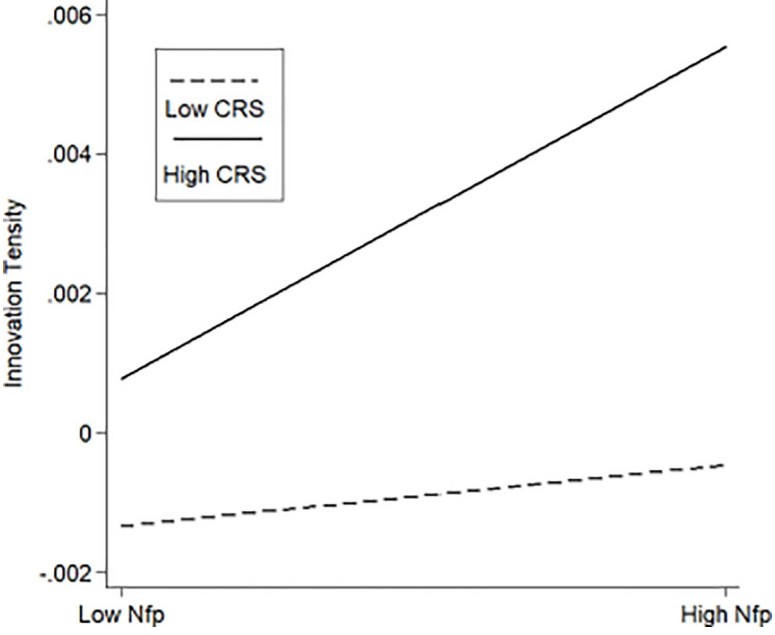

**Fig 4. The moderating effect of the family social responsibility.**

**Table 3. Supplementary regression results with the change in the independent variable.**

| Variable | Model 9 | Model 10 Low control | Model 11 High control | Model 12 Low status | Model 13 High status | Model 14 Low CRS | Model 15 High CRS | Model 16 < 75% | Model 17 > 75% |
|---|---|---|---|---|---|---|---|---|---|
| Nfp2 | 0.0019*** (2.73) | 0.0042* (1.89) | 0.0015** (2.32) | 0.0029** (2.45) | 0.0045*** (2.82) | 0.0025* (1.92) | 0.0055 (1.52) | 0.0039* (1.86) | 0.0058*** (2.93) |
| Gender | 0.0112 (0.18) | 0.0167 (1.18) | 0.0157 (0.14) | -0.0093 (-0.99) | 0.0624 (0.42) | -0.0336 (-0.46) | 0.0367 (0.37) | 0.0294 (0.56) | 0.0203 (0.46) |
| Education | 0.0023*** (2.57) | 0.0028** (2.34) | 0.0030*** (2.93) | 0.0025* (1.84) | 0.0031** (2.12) | 0.0026* (1.79) | 0.0032*** (2.59) | 0.0039** (2.16) | 0.0022* (1.86) |
| Politics | -0.0018 (-0.94) | -0.0035 (-0.32) | -0.0043 (0.39) | 0.0012 (0.31) | -0.0052* (-1.83) | 0.0079** (2.42) | -0.0054** (-2.06) | -0.0036 (0.67) | -0.0015 (-0.88) |
| Ownership | -0.0149* (-1.72) | 0.0167* (1.84) | 0.0083 (0.88) | -0.0134* (-1.87) | -0.0119 (-0.92) | -0.0144 (-0.79) | -0.0113* (-0.712) | -0.0504** (-2.01) | -0.0571 (-1.07) |
| Sale | -0.0284* (-1.78) | -0.0043* (-1.84) | 0.0138 (0.88) | 0.0408 (0.79) | -0.0209 (-0.37) | -0.0584 (-0.48) | 0.0017 (0.33) | 0.0036 (0.64) | -0.0001 (-0.39) |
| Profit | 0.0108 (0.76) | 0.0551 (0.22) | 0.00289* (1.66) | 0.0019 (0.171) | 0.0203 (0.96) | 0.0017 (1.16) | 0.0207 (0.73) | 0.0297 (1.24) | 0.0851 (0.15) |
| Age | 0.0063* (1.36) | 0.0247** (1.98) | -0.0118 (-1.49) | 0.0054 (0.84) | 0.0247* (1.86) | 0.0207** (2.19) | 0.0074 (0.73) | -0.0575* (-1.77) | 0.0167*** (2.75) |
| F_age | 0.0112 (0.14) | -0.0173 (-0.84) | 0.0056* (1.73) | 0.0040 (0.52) | 0.0005 (0.34) | 0.0429 (1.43) | 0.0181 (0.29) | 0.0207 (0.45) | 0.1818 (0.97) |
| Scale | 0.0175 (0.47) | 0.0139 (0.13) | 0.0246 (0.16) | 0.0011 (0.25) | 0.0025 (0.77) | 0.0047 (0.38) | 0.0212 (0.61) | -0.0156 (-0.68) | 0.0761 (0.97) |
| Cons | -0.2844* (-1.75) | -0.1181** (-2.36) | -0.1959* (-1.91) | -0.4754** (-2.03) | -0.5471** (-2.35) | -0.9068** (-2.04) | -0.3628* (-1.85) | 0.1174** (2.42) | -0.5796*** (-3.37) |
| Log likelihood | 2810.77 | 517.82 | 539.79 | 472.53 | 582.79 | 303.32 | 757.26 | 369.12 | 686.59 |
| LR chi2 | 33.91 | 14.95 | 14.49 | 13.22 | 15.07 | 17.88 | 17.45 | 12.68 | 14.28 |
| N | 1098 | 487 | 611 | 550 | 548 | 413 | 685 | 752 | 346 |
| Left | 578 | 251 | 327 | 318 | 260 | 303 | 410 | 520 | 182 |
| Right | 520 | 236 | 284 | 232 | 288 | 110 | 275 | 232 | 164 |
| P-Value | | 0.04 | | 0.07 | | 0.12 | | 0.03 | |

Note: The Z-statistic is shown in parentheses.

* represents $p<0.1$.

** represents $p<0.05$.

*** represents $p<0.01$. Scale and Age are taken by the natural logarithm.

the management of the daily affairs in some Chinese family firms, especially large firms. Hence, it may be unreasonable to assume that these non-family executives make reasonable decisions on R&D investment. For this reason, this study captures Nfp along with the other variable. Specifically, management diversification was determined by investigating the proportion of non-family members in key functions, such as finance, management, technology, and production [99]. From the responses to the questionnaires, the identity of leaders in each key function was judged by knowing whether they were family members, relatives, or friends, and then the proportion of non-family leaders was calculated. This variable is described as Nfp2. Based on this new independent variable, categorical regressions are applied to test the robustness of the moderating effect. The three emotional goals were divided into two categories (Above-average and Below-average), and the Nfp2 coefficients were compared to test the differences.

The results of the robustness test were shown in Table 3. The signs of the coefficients for the independent variables in Model 9 are consistent with the hypotheses proposed, but not all of these coefficients are significant. In Models 10 to 15, the average of three kinds of emotional goal preferences was used as the boundary to the regression tests. Following the method of

unrelated regression, the difference between the coefficients of Nfp2 in each group is signifi-
cant ($P = 0.04$, $P = 0.07$), which indicates that the positive effect of executive diversification on
R&D investment is weaker when the propensity of the family for corporate control is strong.
However, the effect is the opposite in terms of the status perception of the leaders. Although
the difference of the coefficients in the group of CRS is not significant ($P = 0.12$), this does not
mean that the test is invalid. Two factors may account for this result: (1) As can be seen from
Fig 3, the moderating effect of social responsibility is not particularly strong. (2) It inevitably
leads to selective bias when grouping according to the mean value of social responsibility. This
is because only in the case of strong social responsibility, the moderating effect of social
responsibility will be significant, while the difference between above-average levels and below-
average levels in the sample is not obvious. Further robustness checks are done by using three-
quarters of the data as a limit when grouping on the social responsibility. The P-value is 0.05 in
this case. These results confirm Hypothesis 4 proposed.

## 6. Discussion

This research mainly examined the effect of non-family members in TMTs on innovation
investment. The results showed that the involvement of non-family forces in Chinese family
businesses is widespread. These non-family members have creative thinking and multiple per-
spectives that broaden the knowledge base of the family firms. As the introduction of outsiders
accelerates management diversification, it is easy to break the rigid stereotypes for risk aver-
sion [18]. Although this may lead to management conflicts between family members and non-
family members, the long-term orientation of intergenerational transition can effectively
resolve this conflict according to the agency theory. Moreover, increased innovation capabili-
ties will give firms a competitive advantage and leave superior assets to the next generation,
thus marking an inclusive environment. Therefore, suggestions from non-family executives on
innovation strategies are easily accepted [64]. In other words, the management diversity pro-
motes innovation, research, and investment.

As a core theory in the field of family firms, SEW provides insight into the non-economic
behaviors of the firms. However, there are few fine-grained classifications in the research on
strategic decisions of family firms [22]. Although SEW explained the differences in behaviors
between family and non-family enterprises, these emotional goals also some distinctions
among family firms. Combined with the development status of family firms in China, this
paper provided an explanation for the diversified behavior of family firms through the changes
in the concept of SEW in different dimensions, as well as the thinking and behavior of family
members. In addition, the classification of emotional goals into three dimensions of control
and influence, status perception, and social responsibility can effectively provide a reference
for subsequent research, which also helps to reconcile the ambiguity related to TMTs behavior
in the existing literature.

This study fully considered the heterogeneity and the characteristics of the Chinese family
firms and explored the root causes of management conflicts from the perspective of SEW. The
research helps us to understand how the diversification in TMTs affects the innovation invest-
ment under different emotional goals preferences of the leaders of the Chinese family firms.
The results showed that the heterogeneous behaviors of family firms are caused by their differ-
ent emotional goals preferences. As predicted by Hypothesis 2, families with strong corporate
control goals tend to generate short-term orientation, which intensifies the conflict between
family members and non-family members and weakens the contribution of non-family execu-
tives [67], thus inhibiting R&D activities. However, from the dimensions of status perception
and social responsibility, the optimism and the privilege conferred by status could increase the

risk tolerance of the firms when the family leaders have high-status perceptions [75]. Not to mention that the positive sense of social responsibility embodies a win-win corporate culture, which is more likely to obtain external recognition and attract talent. These factors further strengthen the role of diversified management in promoting innovation investment.

## 7. Conclusion

As the main competitors in the modern market, family firms' willingness to invest in R&D determines the technological orientation in the market to a great extent. The involvement of non-family members in TMTs will enable the family firms to make accurate judgments when identifying or pursuing creative and innovative opportunities. This study examined the R&D investment propensity of family firms from the perspective of non-family members' participation in the TMTs. Based on the upper echelons theory, this paper argued that non-family members in the TMTs can bring diversified management capabilities to the family firm, enhance the innovation consciousness of the family firms, and thus significantly increase the R&D investment of the family firms. Further, based on the existing mainstream socioemotional wealth theory, this study took the non-economic goals that influence strategic decisions in family firms into the analysis framework. What's more, combined with the characteristics of Chinese family firms, this paper divided non-economic goals into three dimensions: control and influence, status perception, and social responsibility, and analyzed the moderating effects of these non-economic goals. The empirical results indicated that the preference for control and influence among family members weakens the positive effect of non-family managers on R&D investment, the preference for status perception and social responsibility strengthens the positive effect.

These results contrast with the findings of previous scholars who suggest that the goal of controlling leads to a stronger desire by family members towards the continuity of the family business, and could sustain innovation towards the use of the firms' resources from Spanish family firms [100]. Also, Yoo and Sung [24] argued that outside directors are not effective for promoting R&D intensity, based on the empirical result of Korean firms (1998 to 2005). In this situation, it is believed that heterogeneous conclusions are drawn in different countries, which may be related to the cultural environment of the countries. Although this study regarded China firms as a sample and explained the influence of non-family power on innovation investment in TMTs based on Chinese culture. It is necessary to further combine regional characteristics and the category or scale of firms to conduct a pointed analysis [29] and refine the research objects to reach a more accurate conclusion. Then, although this paper analyzed the possible behaviors of family members in innovation decision-making, it does not broadly focus on all aspects related to non-family executives based on preferences on different affective goal dimensions. Non-family members, senior executives may be influenced by several factors, such as tenure, power status, and even their own personality, which may also have an impact on innovation-related decisions [101]. Finally, in terms of the division of emotional goals, the study is only based on the characteristics of Chinese family businesses. In future research, the classification of emotional goals cannot only be limited to the three methods mentioned in this study, and it could be divided according to different industries and regions [96]. Specifically, Later studies will benefit from a further in-depth analysis by combining psychological methods with the family structures of different family firms.

## Supporting information

**S1 Dataset. The data set used in this article for data analysis and results.**
(XLSX)

## Author Contributions

**Conceptualization:** Yujia Shao, Dechun Huang.

**Data curation:** Lelin Lv, Jie Yu.

**Formal analysis:** Dechun Huang.

**Funding acquisition:** Dechun Huang.

**Investigation:** Lelin Lv, Jie Yu.

**Methodology:** Yujia Shao.

**Project administration:** Yujia Shao, Dechun Huang, Lelin Lv.

**Resources:** Jie Yu.

**Software:** Jie Yu.

**Supervision:** Yujia Shao, Dechun Huang, Lelin Lv.

**Validation:** Dechun Huang.

**Visualization:** Dechun Huang, Jie Yu.

**Writing – original draft:** Yujia Shao.

**Writing – review & editing:** Lelin Lv.

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
