## [Decision Letter · Decision Letter 0]

29 Jun 2021

PONE-D-21-15220

The influence of non-family members in top management teams on research and development investment: Evidence from Chinese family firms

PLOS ONE

Dear Dr. Lelin Lv,

Thank you for submitting your manuscript to PLOS ONE. After careful consideration, we feel that it has merit but does not fully meet PLOS ONE’s publication criteria as it currently stands. Therefore, we invite you to submit a revised version of the manuscript that addresses the points raised during the review process.

There are valuable comments from reviewers and I hope you can revise your manuscript following the comments as suggested. 

We look forward to receiving your revised manuscript.

Kind regards,

Wonjoon Kim, Ph.D

Academic Editor

PLOS ONE

Journal Requirements:

2. Please include additional information regarding the survey or questionnaire used in the study and ensure that you have provided sufficient details that others could replicate the analyses. For instance, if you developed a questionnaire as part of this study and it is not under a copyright more restrictive than CC-BY, please include a copy, in both the original language and English, as Supporting Information

3. In your Data Availability statement, you have not specified where the minimal data set underlying the results described in your manuscript can be found. PLOS defines a study's minimal data set as the underlying data used to reach the conclusions drawn

 in the manuscript and any additional data required to replicate the reported study findings in their entirety. All PLOS journals require that the minimal data set be made fully available. For more information about our data policy, please see http://journals.plos.org/plosone/s/data-availability.

"Upon re-submitting your revised manuscript, please upload your study’s minimal underlying data set as either Supporting Information files or to a stable, public repository and include the relevant URLs, DOIs, or accession numbers within your revised

 cover letter. For a list of acceptable repositories, please see http://journals.plos.org/plosone/s/data-availability#loc-recommended-repositories. Any potentially identifying patient information must be fully anonymized.

Important: If there are ethical or legal restrictions to sharing your data publicly, please explain these restrictions in detail. Please see our guidelines for more information on what we consider unacceptable restrictions to publicly sharing data: http://journals.plos.org/plosone/s/data-availability#loc-unacceptable-data-access-restrictions.

 Note that it is not acceptable for the authors to be the sole named individuals responsible for ensuring data access.

Reviewers' comments:

Reviewer's Responses to Questions

**Comments to the Author**

1. Is the manuscript technically sound, and do the data support the conclusions?

Reviewer #1: Yes

Reviewer #2: Yes

2. Has the statistical analysis been performed appropriately and rigorously? 

Reviewer #1: Yes

Reviewer #2: Yes

3. Have the authors made all data underlying the findings in their manuscript fully available?

Reviewer #1: Yes

Reviewer #2: Yes

4. Is the manuscript presented in an intelligible fashion and written in standard English?

Reviewer #1: Yes

Reviewer #2: No

5. Review Comments to the Author

Reviewer #1: You mentioned in the ethics declaration that NA but clearly this study warranted an ethics clearance. Do kindly look into this matter as it involves humans. Also the referencing needs to be aligned with PLOS ONE criteria. Minor typo errors are encountered in the paper. Updating of a few more relevant and up-to-date journals in the Introduction section will be good and needed.

Reviewer #2: I am glad to review and assess this interesting article, entitled, The influence of non-family members in top management teams on research and development investment: Evidence from Chinese family firms . The organization of this article is good and satisfactory. The Introduction section and methodology portions are adequate. I suggest the authors improve the introduction& Methodology section by adding some latest articles' citations to enhance the work quality and also concise this part. Also, Improve the Conclusion part as well.

Overall, the manuscript is a good piece of work. I recommend that authors do a little more work and add the latest literature to support the study, as suggested. The English level is good and smooth, e.g., the language standard, specifically the grammar, of sufficient quality to meet scientific merit for publication. I accept this manuscript after minor revision, as I have recommended.

6. PLOS authors have the option to publish the peer review history of their article (what does this mean?). If published, this will include your full peer review and any attached files.

Reviewer #1: **Yes: **Anantha Raj A. Arokiasamy

Reviewer #2: No

---

## [Author Response · Author response to Decision Letter 0]

13 Jul 2021

Response to comments

Thank you for your letter and the reviewers’ comments concerning our manuscript entitled “The influence of non-family members in top management teams on research and development investment: Evidence from Chinese family firms” (Manuscript ID: PONE-D-21-15220). Those comments are all valuable and very helpful for revising and improving our paper, as well as the important guiding significance to our researches. We have studied comments carefully and have made correction which we hope meet with approval. Revised portion are marked in red in revised paper. The main corrections in the paper and the responds to the reviewer’s comments are as follows:

Section 1: Reviewer's Responses to Questions

(1) Is the manuscript technically sound, and do the data support the conclusions?

Reviewer #1: Yes

Reviewer #2: Yes

Response: Thank you for reviewers' support and responses.

(2) Has the statistical analysis been performed appropriately and rigorously?

Reviewer #1: Yes

Reviewer #2: Yes

Response: Thank you for reviewers' support and responses. 

(3) Have the authors made all data underlying the findings in their manuscript fully available?

Reviewer #1: Yes

Reviewer #2: Yes

Response: Thank you for reviewers' support and responses. 

(4) Is the manuscript presented in an intelligible fashion and written in standard English?

Reviewer #1: Yes

Reviewer #2: No

Response: Thank you for reviewers' support and responses. 

Responses to Reviewer #2: The authors have thoroughly checked the manuscript for grammar to improve the readability of this study. These changes are highlighted in red in text.

Section 2: Response to Comments for Reviewers

Reviewer #1: You mentioned in the ethics declaration that NA but clearly this study warranted an ethics clearance. Do kindly look into this matter as it involves humans. Also the referencing needs to be aligned with PLOS ONE criteria. Minor typo errors are encountered in the paper. Updating of a few more relevant and up-to-date journals in the Introduction section will be good and needed.

Response: Thank you for reviewers' support and responses. First, we have carefully considered the issue of ethical clearance according to reviewer’s suggestion. Therefore, we have revised the selection of ethics declaration in the process of submitting a revised version. Second, we have revised the references to be consistent with PLOS ONE criteria. Also, we have updated some more relevant and up-to-date journals in the full text, especially in the Introduction section. Finally, we have performed a detailed check of the manuscript to improve the readability of this study. These changes above are highlighted in red in text.

(1) Originality: Does the paper contain new and significant information adequate to justify publication? This paper investigates the R&D investment propensity of family firms from the perspective of non-family members’ participation in TMTs. Based on the upper echelons and the socioemotional wealth theory, this paper incorporated the non-economic goals that influence strategic decisions in family firms into the analytical framework. The paper is very interesting and is expected to contribute to the readers in emerging and established markets. The suggestions are also timely and appropriate. Overall, it is a good piece of work. However, I have some suggestions which could help to improve on the quality of your paper especially in your introduction section. Specifically, you need to support your introduction with a few more (e.g., 2-3) up-to-date journal articles as some of them are outdated.

Response: Thank you for reviewers' support and responses. We have updated thirteen journal articles according to reviewer’s suggestion. These changes above are highlighted in red in text. 

(2) Relationship to literature: Does the paper demonstrate an adequate understanding of the relevant literature in the field and cite an appropriate range of literature sources? Is any significant work ignored? The literature review is comprehensive with good perspectives to each of the points and captured the latest works in the related areas. Additionally, the hypotheses development is well written. Well done! May I suggest including a conceptual framework for the reader’s imagination on the build-up of the hypotheses.

Response: Thank you for reviewers' support and responses. We have added a conceptual framework in the Hypotheses section according to reviewer’s suggestion. These changes are highlighted in red in text. The details are as follows:

Based on the characteristics of Chinese family firms, this study examined the impact of importing outsiders into TMTs on the strategic management and innovation goal intention of family firms. In addition, considering the influence of family members on management decisions, this study divided the affective goals of family members into three dimensions: control and influence, status perception, and social responsibility. It also analyzed the moderating effects of these emotional goals on the relationship between the involvement of non-family members in TMT and R&D investment. Fig. 1 shows a visual representation of the core constructs in our study and the proposed relationships among them.

(3) Results: Are results presented clearly and analyzed appropriately? Do the conclusions adequate tie together the other elements of the paper? Yes, the results are presented and analyzed appropriately.

Response: Thank you for reviewers' support and responses.

(4) Implications for research, practice and or society. Does the paper identify clearly between any implications for research, practice and or society? Does the paper bridge the gap between theory and practice? How can the research be used in practice, in teaching, to influence public policy, in research? What is the impact upon society? Are these implications consistent with the findings and conclusion of the paper? They are well written! Well done!

Response: Thank you for reviewers' support and responses.

(5) Quality of communication: Does the paper clearly express its case, measured against the technical language of the field and the expected knowledge of the journal’s readership? Has attention been paid to the clarity of expression and readability, such as sentence structure, jargon use, acronyms, etc. The reviewer spotted some grammatical errors in the paper; please do thorough grammatical checking to the manuscript to improve the readability of this study.

Response: Thank you for reviewers' support and responses. The authors have thoroughly checked the manuscript for grammar to improve the readability of this study. These changes are highlighted in red in text.

(6) References. The references are not in line with PLOS ONE. May I suggest changing it to the required references if the manuscript is accepted for publication.

Response: Thank you for reviewers' support and responses. We have revised the references to be consistent with PLOS ONE criteria.

Reviewer #2: I am glad to review and assess this interesting article, entitled, The influence of non-family members in top management teams on research and development investment: Evidence from Chinese family firms. The organization of this article is good and satisfactory. The Introduction section and methodology portions are adequate. I suggest the authors improve the introduction & Methodology section by adding some latest articles' citations to enhance the work quality and also concise this part. Also, Improve the Conclusion part as well. Overall, the manuscript is a good piece of work. I recommend that authors do a little more work and add the latest literature to support the study, as suggested. The English level is good and smooth, e.g., the language standard, specifically the grammar, of sufficient quality to meet scientific merit for publication. I accept this manuscript after minor revision, as I have recommended. 

Response: Thank you for reviewers' support and responses. In the Introduction and Methodology section, we have added citations to some recent articles to improve the work quality and have simplified these sections. Moreover, in order to enhance the value of considering the country context of our study, we compared findings with those from studies in different countries, which reflects the particularity of family firms in Chinese context. These changes are highlighted in red in text.

Section 3: Response to Additional Requirements

Response: We have revised the manuscript according to the style requirements of PLOS ONE.

(2) Please include additional information regarding the survey or questionnaire used in the study and ensure that you have provided sufficient details that others could replicate the analyses. For instance, if you developed a questionnaire as part of this study and it is not under a copyright more restrictive than CC-BY, please include a copy, in both the original language and English, as Supporting Information

Response: This study did not develop a questionnaire. The original data in the manuscript came from the sample survey database of Chinese private enterprises in 2010 (https://cpes.zkey.cc/index.jsp).

(3) In your Data Availability statement, you have not specified where the minimal data set underlying the results described in your manuscript can be found. 

Response: We have supported our minimal dataset as a supporting Information file, which is from Chinese Private Enterprise Survey (CPES).

(4) Please include captions for your Supporting Information files at the end of your manuscript, and update any in-text citations to match accordingly.

Response: We have added the title of the supporting information files at the end of the manuscript. These changes are highlighted in red in text.

Special thanks for reviewers’ and editor’s comments. We tried our best to improve the manuscript and made some changes in the manuscript. These changes will not influence the content of the paper and the changes were highlighted in red in text.

We appreciate for Editors/Reviewers’ warm work earnestly, and hope that the correction will meet with approval.

Once again, thank you very much for your comments and suggestions.

With all best wishes.

---

## [Editor Report · Decision Letter 1]

12 Aug 2021

PONE-D-21-15220R1

The influence of non-family members in top management teams on research and development investment: Evidence from Chinese family firms

PLOS ONE

Dear Dr. Lelin Lv,

Thank you for submitting your manuscript to PLOS ONE. After careful consideration, we feel that it has merit but does not fully meet PLOS ONE’s publication criteria as it currently stands. Therefore, we invite you to submit a revised version of the manuscript that addresses the points raised during the review process.

Thank you for your revision and the current manuscript had substantial progresses. However, there are still some issues that authors need to clarify and updates in the revision. First, as one reviewer pointed out, this study involves humans (survey). Please clarify this issue or provide approval for human subjects research by an institutional review board (IRB) or equivalent ethics committee(s) if necessary. Second, reference needs to be updated reflecting the latest literature. There are typo and errors in the paper. Please proof edit the manuscript before submission.

We look forward to receiving your revised manuscript.

Kind regards,

Wonjoon Kim, Ph.D

Academic Editor

PLOS ONE
---

## [Author Response · Author response to Decision Letter 1]

13 Aug 2021

Response to comments

Thank you for your letter and the comments concerning our manuscript entitled “The influence of non-family members in top management teams on research and development investment: Evidence from Chinese family firms” (Manuscript ID: PONE-D-21-15220). The main corrections in the paper are as follows:

(1) First, as one reviewer pointed out, this study involves humans (survey). Please clarify this issue or provide approval for human subjects research by an institutional review board (IRB) or equivalent ethics committee(s) if necessary.

Response: Thank you for your support and response. We have revised the selection of ethics declaration in the process of submitting a revised version and added this statement at the beginning of the Methodology section of manuscript. These changes are highlighted in red in text. The details are as follows: The names of the institutional review board that approved the questionnaire survey are the Institute of Sociology Chinese Academy of Social Sciences and All-China Federation of Industry and Commerce, which issued a statement approving this research (http://ssm.cssn.cn/sjzy/201609/t20160914_3202846.shtml). Moreover, the survey data was analyzed anonymously.

(2) Second, reference needs to be updated reflecting the latest literature.

Response: Thank you for your support and response. We have updated seventeen journal articles according to the suggestion. These changes are highlighted in red in text. The details are as follows:

2. Calabro A, Vecchiarini M, Gast J, Campopiano G, De Massis A, Kraus S. Innovation in Family Firms: A Systematic Literature Review and Guidance for Future Research. International Journal of Management Reviews. 2019;21(3):317-355. doi: 10.1111/ijmr.12192.

4. Classen N, Carree M, Van Gils A, Peters B. Jiang F, Shi W, Zheng X. Board chairs and R&D investment: Evidence from Chinese family-controlled firms. Journal of Business Research. 2020;112(5):109-118. doi: 10.1016/j.jbusres.2020.02.026.

6. Dieguez-Soto, J, Martinez-Romero, MJ. Family Involvement in Management and Product Innovation: The Mediating Role of R&D Strategies. Sustainability. 2019;11(7):1-24. doi: 10.3390/su11072162.

7. Zulfiqar M, Zhang R, Khan N. Behavior Towards R&D Investment of Family Firms CEOs: The Role of Psychological Attribute. Psychology Research and Behavior Management. 2021;14:595-620. doi: 10.2147/PRBM.S306443.

19. Wu T, Wu YJ, Tsai H, Li Y. Top Management Teams' Characteristics and Strategic Decision-Making: A Mediation of Risk Perceptions and Mental Models. Sustainability. 2017;9(12):1-15. doi: 10.3390/su9122265.

21. De Massis A, Eddleston KA, Rovelli P. Entrepreneurial by Design: How Organizational Design Affects Family and Non-family Firms' Opportunity Exploitation. Journal of Management Studies. 2021;58(1):27-62. doi: 10.1111/joms.12568.

25. Prokop V, Stejskal J, Klimova V, Zitek V. The role of foreign technologies and R&D in innovation processes within catching-up CEE countries. PLoS One. 2021;16(4):e0250307. doi: 10.1371/journal.pone.0250307.

33. Yang B, Nahm A, Song Z. Succession, political resources, and innovation investments of family businesses: Evidence from China. Managerial and Decision Economics. 2021. doi: 10.1002/mde.3385.

35. Ge Y, Kong X, Dadilabang G, Ho K. The effect of Confucian culture on household risky asset holdings: Using categorical principal component analysis. International Journal of Finance & Economics. 2021. doi: 10.1002/ijfe.2452.

39. Carmen Diaz-Fernandez M, Rosario Gonzalez-Rodriguez M, Simonetti B. The role played by job and non-job-related TMT diversity traits on firm performance and strategic change. Management Decision. 2016;54(5):1110-1139. doi: 10.1108/md-10-2015-0464.

78. Newbert S, Craig JB. Moving Beyond Socioemotional Wealth: Toward a Normative Theory of Decision Making in Family Business. Family Business Review. 2017;30(4):339-346. doi: 10.1177/0894486517733572. 

79. El Akremi A, Gond J-P, Swaen V, De Roeck K, Igalens J. How Do Employees Perceive Corporate Responsibility? Development and Validation of a Multidimensional Corporate Stakeholder Responsibility Scale. Journal of Management. 2018;44(2):619-657. doi: 10.1177/0149206315569311.

82. Li X. The effectiveness of internal control and innovation performance: An intermediary effect based on corporate social responsibility. PLoS One. 2020;15(6):e0234506. doi: 10.1371/journal.pone.0234506. 

83. Alpkan L, Gülsoy T. Introduction to the Special Issue on Strategic Management in Emerging Economies: Innovation, Corporate Social Responsibility, and Sustainable Management. International Journal of Innovation and Technology Management. 2019;16(4):1-4. doi: 10.1142/S0219877019020012.

93. Mehrabi H, Coviello N, Ranaweera C. When is top management team heterogeneity beneficial for product exploration? Understanding the role of institutional pressures. Journal of Business Research. 2021;132:775-786. doi: 10.1016/j.jbusres.2020.10.057.

95. Salah OH, Yusof ZM, Mohamed H. The determinant factors for the adoption of CRM in the Palestinian SMEs: The moderating effect of firm size. PLoS One. 2021;16(3):e0243355. doi: 10.1371/journal.pone.0243355.

101. Lee K, Makri M, Scandura T. The Effect of Psychological Ownership on Corporate Entrepreneurship: Comparisons Between Family and Nonfamily Top Management Team Members. Family Business Review. 2019;32(1):10-30. doi: 10.1177/0894486518785847.

(3) There are typo and errors in the paper. Please proof edit the manuscript before submission.

Response: Thank you for your support and response. We have performed a detailed check of the manuscript to improve the readability of this study. These changes are highlighted in red in text.

Special thanks for reviewers’ and editor’s comments. We tried our best to improve the manuscript and made some changes in the manuscript. These changes will not influence the content of the paper and the changes were highlighted in red in text.

We appreciate for Editors/Reviewers’ warm work earnestly, and hope that the correction will meet with approval.

Once again, thank you very much for your comments and suggestions.

With all best wishes.

---

## [Decision Letter · Decision Letter 2]

22 Sep 2021

The influence of non-family members in top management teams on research and development investment: Evidence from Chinese family firms

PONE-D-21-15220R2

Dear Dr. Lelin Lv,

We’re pleased to inform you that your manuscript has been judged scientifically suitable for publication and will be formally accepted for publication once it meets all outstanding technical requirements.

Kind regards,

Wonjoon Kim, Ph.D

Academic Editor

PLOS ONE

Additional Editor Comments (optional):

Reviewers' comments:

Reviewer's Responses to Questions

**Comments to the Author**

1. If the authors have adequately addressed your comments raised in a previous round of review and you feel that this manuscript is now acceptable for publication, you may indicate that here to bypass the “Comments to the Author” section, enter your conflict of interest statement in the “Confidential to Editor” section, and submit your "Accept" recommendation.

Reviewer #1: All comments have been addressed

Reviewer #2: All comments have been addressed

2. Is the manuscript technically sound, and do the data support the conclusions?

Reviewer #1: Yes

Reviewer #2: Yes

3. Has the statistical analysis been performed appropriately and rigorously? 

Reviewer #1: Yes

Reviewer #2: Yes

4. Have the authors made all data underlying the findings in their manuscript fully available?

Reviewer #1: Yes

Reviewer #2: Yes

5. Is the manuscript presented in an intelligible fashion and written in standard English?

Reviewer #1: Yes

Reviewer #2: Yes

6. Review Comments to the Author

Reviewer #1: Great, all comments addressed well. The paper has been revised according to all the comments raised by the reviewers and its deemed good for publication.

Reviewer #2: I am glad to review and assess this exciting article, entitled "The influence of non-family members in top management teams on research and development investment: Evidence from Chinese family firms" . Hence I recommend the acceptance of this paper.

7. PLOS authors have the option to publish the peer review history of their article (what does this mean?). If published, this will include your full peer review and any attached files.

Reviewer #1: **Yes: **Dr. Anantha Raj A. Arokiasamy

Reviewer #2: No

---

## [Editor Report · Acceptance letter]

30 Sep 2021

PONE-D-21-15220R2 

The influence of non-family members in top management teams on research and development investment: Evidence from Chinese family firms 

Dear Dr. Lv:

I'm pleased to inform you that your manuscript has been deemed suitable for publication in PLOS ONE. Congratulations! Your manuscript is now with our production department. 

Kind regards, 

on behalf of

Dr. Wonjoon Kim 

Academic Editor

PLOS ONE